The effect of 2,4-dichlorophenoxyacetic acid on the production of oat (Avena sativa L.) doubled haploid lines through wide hybridization

Juzoń Katarzyna k.juzon@ifr-pan.edu.pl
Warchoł Marzena
Dziurka Kinga
Czyczyło-Mysza Ilona Mieczysława
Marcińska Izabela
Skrzypek Edyta
Instytut Fizjologii Roślin im. Franciszka Górskiego PAN , Kraków , Polska
Liu Xin
Electronic publication date: 2022 Jan 31
Publication date: 2022
Volume: 10
Electronic Location ID: e12854
Received 2021 Aug 18; Accepted 2022 Jan 7
Copyright: ©2022 Juzoń et al.
Copyright year: 2022
Copyright holder: Juzoń et al.
License: This is an open access article distributed under the terms of the Creative Commons Attribution License, which permits unrestricted use, distribution, reproduction and adaptation in any medium and for any purpose provided that it is properly attributed. For attribution, the original author(s), title, publication source (PeerJ) and either DOI or URL of the article must be cited.
License URL: https://creativecommons.org/licenses/by/4.0/

Keywords: Avena sativa L.; 2,4-dichlorophenoxyacetic acid (2,4-D); Doubled haploid (DH) lines; Wide crossing; Zea mays L.

Funding: The Ministry of Agriculture and Rural Development, Poland HORhn-801-1/13 The work was supported by the Ministry of Agriculture and Rural Development, Poland, grant No. HORhn-801-1/13. The funders had no role in study design, data collection and analysis, decision to publish, or preparation of the manuscript.

==============================
Background

Development of new cultivars is one of the vital options for adapting agriculture to climate change, and the production of doubled haploid (DH) plants can make a significant contribution to accelerating the breeding process. Oat is one of the cereals with particular health benefits, but it unfortunately still remains recalcitrant to haploidization. Our previous studies have clearly demonstrated that post-pollination with hormone treatment is a key step in haploid production through wide hybridization and indicated it as the most effective method for this species. Therefore, we subsequently addressed the problem of the influence of 2,4-dichlorophenoxyacetic acid (2,4-D) concentration on consecutive stages of DH production.

Methods

Twenty-nine genotypes were tested, 9,465 florets were pollinated with maize pollen 2 days after emasculation and then treated with 2,4-D at 50 mg/L and 100 mg/L.

Results

The applied treatments did not reveal any differences in the number of obtained haploid embryos. However, almost twice as many haploid plants formed on MS medium after applying a higher auxin concentration and 20% more successfully acclimatized. Moreover, 100 mg/L 2,4-D treatment resulted in twice as many DH lines that produced almost three times more seeds compared to 50 mg/L treatment. Nevertheless, the results have confirmed the existence of strong genotypic variation, which may significantly limit the development of an effective and economically feasible method that could be incorporated into breeding programs.

Introduction

Oat (Avena sativa L.) constitutes the sixth most important cereal crop (Bekele et al., 2018; Hu et al., 2020) and remains a highly valued component of the human diet associated with dietary fiber such as β-glucan, functional protein, lipid and starch components, phytochemicals (Butt et al., 2008; Rasane et al., 2015) as well as phenolic compounds with antioxidant activity (Gray et al., 2002). All these features make oat products highly recommended in gastrointestinal ailments and neoplastic conditions, classified as civilization diseases. However, oat production and its quality, like that of other cereal crops, strongly depends on climatic conditions. Currently, our population is facing the problem of long-term changes in the state of the climate (IPCC, 2014). These alterations are believed to influence many aspects of our life, but their effects on agricultural production seem to be particularly severe. Despite the various obstacles to agricultural production, consumers will continue to demand tasty, convenient, healthy and safe food (Henry, 2020). One possible solution to counteract the negative impact of climate change is the development of new plant genotypes adapted to highly variable environmental conditions and pest infestations (Korres et al., 2016; Henry, 2020).

Doubled haploid (DH) technology enables the instantaneous development of homozygous lines, which is a major breakthrough in accelerating cultivar development (Dunwell, 2010), instead of 6-10 generations of inbreeding by selfing or sib-crossing (Prigge et al., 2012). DHs are produced by androgenesis, gynogenesis, wide hybridization or interspecific crosses (Ferrie et al., 2014). This technology for practical breeding is available in barley (Jacquard et al., 2006), wheat (Niu et al., 2014), and maize (Chaikam et al., 2019), in which thousands of DHs are produced annually using routine methods. However, in rye (Immonen & Tenhola-Roininen, 2003), triticale (Immonen & Robinson, 2000), potato (Rokka, Pietila & Pehu, 1996) and cabbage (Hansen, 2003), DH technologies are less advanced. Unfortunately, despite countless health benefits, oat is considered as a species recalcitrant to haploidization (Sidhu et al., 2006; Marcińska et al., 2013; Skrzypek et al., 2016; Dziurka et al., 2019). There are certain barriers that operates at pre and post fertilization phases which hinders the effectiveness of wide crossing of oat with maize. Firstly, oat endosperm usually fails to develop in seeds and the embryos must be rescued by transferred on culture medium to allow them to grow under optimum culture conditions. Therefore, selection of regeneration medium is the most critical step for the continued growth of the embryo. Moreover, all authors describe a very strong dependence of the development of haploid plants on the tested genotypes. So far, the efficiency of DH production in wheat reaches about 20% haploid per florets while in oat is less than one percent (Sidhu et al., 2006). Oat haploids have been obtained by in vitro microspore/anther culture (Rines, 1983; Sun, Lu & Sõndahl, 1991; Kiviharju, Moisander & Laurila, 2005; Ponitka & Ślusarkiewicz Jarzina, 2009; Ferrie et al., 2014) as well as wide hybridizations (Rines, Davis & Busch, 1991; Rines, 2003; Sidhu et al., 2006; Marcińska et al., 2013; Nowakowska et al., 2015), which is commonly indicated as the most effective method for this species (Rines, 2003; Sidhu et al., 2006; Ishii et al., 2013; Marcińska et al., 2013; Nowakowska et al., 2015; Warchoł et al., 2018). Therefore, wide crossing with maize (Zea mays L.) is most commonly applied in the commercial production of oat doubled haploid (DH) lines (Skrzypek et al., 2016), and no albino plants are produced among regenerants (Rines & Dahleen, 1990).

Although numerous studies on the optimization of oat DH production have been published so far (Rines, 2003; Sidhu et al., 2006; Marcińska et al., 2013; Nowakowska et al., 2015; Warchoł et al., 2018; Warchoł et al., 2019), the efficiency of this process is still disappointing, especially in the context of its use in breeding programs (Sidhu et al., 2006). Many efforts have been made to increase its effectiveness and the work have focused, e.g., on (i) the species of the pollen donor (Rines, 2003; Sidhu et al., 2006; Ishii et al., 2013; Nowakowska et al., 2015), (ii) applied media (Rines & Dahleen, 1990; Kynast et al., 2001; Sidhu et al., 2006; Ishii et al., 2013; Nowakowska et al., 2015; Warchoł et al., 2018), (iii) type and concentration of supplemented carbohydrates (Rines et al., 1997; Kynast & Riera-Lizarazu, 2011; Ishii et al., 2013; Marcińska et al., 2013; Niu et al., 2014; Nowakowska et al., 2015; Noga et al., 2016; Skrzypek et al., 2016; Warchoł et al., 2018) or (iv) light intensity during in vitro cultures (Skrzypek et al., 2016). Wide crossing, involving fertilization with alien pollen, reduces embryo viability due to the absence of proper endosperm development. Therefore, the application of plant growth regulators (PGRs), which follows floret emasculation, is one of the crucial aspects in obtaining haploid embryos. 2,4-D is the auxin analogue the most widely used to control organ regeneration, callus induction or somatic embryogenesis induction (Wedzony et al., 2015). Studies have shown that 2,4-D plays a central role in early and post-embryogenic plant development and mediates important steps during the early embryo patterning formation of zygotic embryogenesis (Nic-Can & Loyola-Vargas, 2016). Moreover, exogenously applied auxins change the level of endogenous auxins such as IAA and modify its metabolism inside the cell (Vondráková et al., 2016). The increment in the endogenous IAA regulates the expression of a great number of transcription factors, several of them related to stress. Exogenous auxin seems to be required for the establishment of a normal embryonic symmetry at the globular and early transition stages embryos since both stages were affected by manipulation of the auxin level or distribution. This dependence was confirmed by immunohistological analysis of IAA in isolated zygotic and somatic embryos of cereals cultured in vitro (Fischer & Neuhaus, 1996; Przetakiewicz, Orczyk & Nadolska-Orczyk, 2003; Forestan, Meda & Varotto, 2010; Kruglova, Seldimirova & Zinatullina, 2020). These data emphasize that the polar transport of auxin is essential for the formation of embryonic patterns and the distribution of this hormone is at all stages of embryo development, especially with the formation of bilateral symmetry in embryos. In DH line production, developed zygotes have a very low viability, and most of them abort during the initial stages of development. The application of auxins induces ovary enlargement and enhances the growth of the haploid embryos to a stage that enables their culture onto nutrient media (Laurie & Reymondie, 1991) (12–16 days post-pollination in wheat) (Mahato & Chaudhary, 2019). There is a general agreement on the need for applying synthetic auxins in DH line production as a means of recovering haploid embryos especially to applying them following pollination. The benefits of auxins is inducing rapid vacuolization and hydrolyzation of the embryo sac which, in turn, restrains fertilization (Matzk, 1991). Various auxin analogues (picloram; dicamba; 2,4-dichlorophenoxyacetic acid (2,4-D), gibberellic acid) (reviewed in Sidhu et al. (2006)) were tested for their ability to induce caryopsis and embryo formation, but no statistically significant differences between the applied substances or significant growth regulator × genotype interactions have been established so far (Warchoł et al., 2018). Marcińska et al. (2013) reported that dicamba (3,6-dichloro-2-methoxybenzoic acid) treatment increased the size of the ovaries; however, 2,4-D application turned out to be more effective in converting embryos to haploid plants and obtaining DHs (Warchoł et al., 2018). Furthermore, haploid production efficiency is also affected by the concentration, time and hormone application method (Mahato & Chaudhary, 2019).

Our previous long-term study have led to the development of a methodology for obtaining oat DH that assumed that pollination on the 2nd day after floret emasculation, and auxin treatment in the following 2 days were the most effective. In this study, we decided to investigate post-pollination application of different 2,4-D concentrations in order to improve the efficiency of oat double haploid production by wide hybridization with maize. The purpose of this experiment was to demonstrate the influence of various auxin concentrations on each step of the process, i.e., haploid embryo formation, regeneration of haploid and DH plants as well as their fertility. The high complexity of this process demonstrates that there is still a need to explore the mechanisms underlying DH production and all factors conditioning its effectiveness.

Material and Methods

Plant material

Twenty-nine oat (Avena sativa L.) genotypes (F1 progeny) were used in the study. Oat seeds were obtained from Danko Plant Breeding Ltd. (DC 11003, DC 11021, DC 11027, DC 11033, DC 11040, DC 11125, DC 11142, DC 11146, DC 11164, DC 11209, DC 11221 and DC 11244), Małopolska Plant Breeding Ltd. HBP Polanowice (POB 14/2013, POB 15/2013, POB 19/2013, POB 20/2013, POB 21/2013 and POB 38/2013) and Plant Breeding Strzelce Ltd., PBAI Group (STH 2.3602, STH 2.3618, STH 2.3619, STH 2.3621, STH 2.3626, STH 2.3628, STH 2.3644, STH 2.3646, STH 2.3659, STH 2.3668 and STH 2.9156). A mixture of three maize (Zea mays L. var. saccharata) genotypes was used as a pollen donor: MPC4, Dobosz and Wania according to Skrzypek et al. (2016). Oat and maize plants were grown in the glasshouse under natural light conditions (photosynthetic active radiation (PAR) of 800 µmol m−2 s−1 and 16 h light/8 h dark) as well as optimal temperature for each species (21/17 °C day/night and 21-28/17 °C day/night for oat and maize respectively).

Haploid plant production

Two days after manually emasculation (Fig. 1A) florets were pollinated with a fresh maize pollen (collected at 15-min intervals) using a brush (Fig. 1B). The next day, one drop of 50 mg/L or 100 mg/L 2,4-D water solution was applied to each oat pistil (Fig. 1C). Three weeks later, enlarged ovaries (caryopses without endosperms) were surface-sterilized in 70% v/v ethanol (1 min), 2.5% calcium hypochlorite (7 min), 0.1% mercuric chloride (1 min), washed three times with sterile water and then the embryos were isolated (Fig. 1D), transferred onto 190-2 medium (Wang & Hu, 1984) and cultured at 21 ± 2 °C, at the 16 h light at the intensity 60 µmol m−2 s−1. The developed haploid plants were grown on MS medium (Murashige & Skoog, 1962) (Fig. 1E), then transferred to a moist perlite (PPUH Perlit Polska Ltd., Puńców, Poland) (Fig. 1F) and subsequently to soil (Fig. 1G) for acclimatization to ex vitro conditions. DH line plants were planted individually into 3 dm3 pots with a diameter of 16 cm filled with soil composed of horticultural soil (Ekoziem, Jurków, Poland) and sand (2:1 v/v). Plants’ acclimatization was performed in the glasshouse condition described above. The seeds were obtained after 5–6 months. To show the efficiency of this method, the number of obtained haploid embryos per one hundred of pollinated florets was calculated.

Figure 1 Diagram of the production of oat doubled haploid lines through the wide crossing method.

(A) Floret emasculation, (B) floret pollination with maize pollen, (C) 2,4-D application to the pistil, (D) embryo isolation, (E) in vitro culture of embryos on MS medium and their conversion into haploid plants, (F) haploid plants grown in perlite, (G) haploid plant acclimatization in soil, (H) chromosomes doubling by colchicine treatment, (I) DH plant grown in soil until maturity.

Chromosome doubling

To double the number of chromosomes, when oat seedlings were at 4–5 leaf stage, colchicine treatment (0.1% colchicine solution) (Fig. 1H) was carried out according to (Warchoł et al., 2018) for 7.5 h at 25 °C and 80–100 µmol m−2 s−1 light intensity. Next, the plant roots were rinsed with running water for 48 h, and then planted singly into pots (Fig. 1I) for further growth and maturation under greenhouse conditions (temperature 21/17 °C day/night in natural solar light). Plant ploidy level was measured before and one month after colchicine treatment (Fig. 2) using a MACS Quant flow cytometer (Miltenyi Biotec GmbH, Bergisch Gladbach, Germany), equipped with an air-cooled laser (488 nm) and MACSQantify™ software. The young leaves (10–15 mg) were placed in a 60-mm glass Petri dish with 1 ml of the modified PBS buffer (Sambrook, Fritsch & Maniatis, 1989) and chopped with a razor blade in order to release nuclei and then filtered with a 30- µm nylon mesh filter (Miltenyi Biotec GmbH, Bergisch Gladbach, Germany). The nuclei suspension was stained with 20 µl of propidium iodide solution. As a control, oat plants derived from the seeds of known diploid DNA content were used.

Figure 2 Flow cytometry histograms of oat plants: (A) control 2n –cv. Bingo, (B) haploid 1n –DC11142 before colchicine treatment, and (C) doubled haploid 2n –DC11142 after colchicine treatment.

All reagents used in the experiment were obtained from Sigma-Aldrich® (Sigma Aldrich, Darmstadt, Germany).

Statistical analysis

Data analysis was performed using two-way ANOVA implemented in the statistical package STATISTICA 13.0 (TIBCO Software Inc., Palo Alto, CA, USA). Differences between treatments were considered significant at p ≤ 0.005.

Results

The analysis of variance showed significant differences in the number of haploid plants obtained on MS medium relative to the tested genotype, auxin treatment (50 mg/L or 100 mg/L) and their interaction (Table 1). All examined factors were not significant with respect to the number of haploid embryos and DH lines produced.

Table 1 Two way analysis of variance in oat doubled haploid production using the wide hybridization method, showing the significance of genotype, auxin treatment and their interaction with measured trait.

Trait	Source of variance	SS	df	MS	F	p	
Number of haploid embryos/
100 emasculated florets	Genotype	4823.98	28 (g-1)	172.285	58.618	2 × 10−3ns	
	Auxin	18.02	1 (a-1)	18.023	218.000	1 × 10−3ns	
	Genotype × Auxin	2645.98	28 (a-1) ×(g-1)	94.499	49.123	4 × 10−3ns	
	Residual error	1 ×10−3	116 (a ×g-1) ×(n-1)	0.000			
Number of haploid plants on
MS medium/100 emasculated
florets	Genotype	184.29	33 (g-1)	5.585	63.293	1 ×10−5∗	
	Auxin	29.82	1 (a-1)	29.824	338.000	3 ×10−5∗	
	Genotype × Auxin	85.18	33 (a-1) ×(g-1)	2.581	29.253	4 ×10−5∗	
	Residual error	12.00	136 (a ×g-1) ×(n-1)	0.088			
Number of DH lines/
100 emasculated florets	Genotype	176.69	28 (g-1)	6.310	61.567	1 × 10−2ns	
	Auxin	25.03	1 (a-1)	25.034	325.790	2 × 10−3ns	
	Genotype × Auxin	82.97	28 (a-1) ×(g-1)	2.963	28.240	3 × 10−3ns	
	Residual error	2 ×10−3	116 (a ×g-1) ×(n-1)	0.000			
Notes.

SS, sum of squares, df, degrees of freedom, MS, mean squares, ns, not significant.

∗ p ≤ 0.005.

Effect of 2,4-D concentration on oat haploid embryo formation

Nearly 9,500 oat florets from 29 genotypes were emasculated and pollinated with maize pollen, and then over 8,500 enlarged ovaries (some ovaries did not enlarge—Fig. 3A) were isolated (Table 2, Figs. 3B, 3C). In total, 619 haploid embryos were obtained (Figs. 3D, 3E) Their number between the tested genotypes and 2,4-D concentrations ranged from 1 to 31. Two genotypes, DC 11003 and DC 11221, developed the highest number of haploid embryos, while genotypes POB 38/2013 and STH 2.9156 the lowest, both after the application of 50 mg/L 2,4-D. Moreover, the haploid-embryo-per-floret index reached the highest values (over 20%) at the lower auxin concentration. Most of the tested genotypes showed values ranging from 5 to 10% –16 genotypes (55%) at 50 mg/L 2,4-D, while 22 genotypes (76%) at 100 mg/L 2,4-D.

Figure 3 Oat doubled haploids production by wide crossing with maize.

(A) Non-enlarged ovary, (B) enlarged ovary after 50 mg/L2, 4-D treatment, (C) enlarged ovary after 100 mg/L 2,4-D treatment, (D) haploid embryo inside ovary, (E) haploid embryo isolated from ovary, (F) germinated haploid embryo, (G) developed haploid plant, (H) haploid plants acclimated to the natural conditions before colchicine treatment, (I) plants after colchicine treatment, (J) maturating DH plants in the greenhouse, (K) panicles of DH lines DC 11003, (L) panicles of DH lines POB 19/2013, (M) panicles of DH lines STH 2. 3619. Photo credits: Marzena Warchoł and Edyta Skrzypek.

Effect of 2,4-D concentration on haploid plant development

However, obtaining haploid embryos as the first step of this method did not guarantee the high final efficiency of obtaining haploid plants. Despite the high number of isolated embryos (619), only 12.3% (76) germinated within 1-3 weeks of culturing (Fig. 3F, Table 2). Embryos from approximately a quarter of the studied genotypes (DC 11164, POB 14/2013, POB 15/2013, POB 20/2013, POB 38/2013, STH 2.3659 and STH 2.9156) did not germinate at all or died during development and, in consequence, failed to regenerate haploid plants, regardless of the applied auxin dose. Nevertheless, a clear difference between the two applied 2,4-D concentrations was noticed at this step: 50 mg/L 2,4-D resulted in an average of 27 haploid plants (8.5%), while 49 haploids (16.3%) were obtained at 100 mg/L 2,4-D (Fig. 3G). Although the highest number of haploid plants (five haploids in genotype DC 11221) was recorded at the lower 2,4-D concentration, this treatment for most of the studied genotypes did not seem to be as favorable as the higher auxin dose. Embryos of 17 genotypes did not regenerate into plants at 50 mg/L 2,4-D, while embryos of only seven genotypes did not undergo further development after 100 mg/L 2,4-D treatment. Moreover, the higher auxin concentration allowed to obtain four haploid plants in four studied genotypes (DC 11003, DC 11142, DC 11221 and STH 2.3619), and three haploid plants in five genotypes (DC 11146, DC 11209, POB 21/2013, STH 2.3618 and STH 2.3644).

Table 2 The influence of 2,4-D concentration (50 mg/L or 100 mg/L) on the efficiency of haploid embryo and DH plant production using the wide crossing method.

Genotype	Pollinated florets	Isolated ovaries	Haploid embryos	Haploid embryos/ florets [% ± SE]	Haploid plants on MS medium	Plants after colchicine treatment	
	50 mg	100 mg	50 mg	100 mg	50 mg	100 mg	50 mg	100 mg	50 mg	100 mg	50 mg	100 mg	
DC 11003	139	139	131	135	31	15	25.8 ± 3.06	11.0 ± 0.83	2	4	2	4	
DC 11021	197	124	193	121	15	12	13.9 ± 3.30	9.4 ± 0.50	3	2	3	2	
DC 11027	302	133	268	137	20	10	8.6 ± 1.46	7.5 ± 0.73	1	1	1	1	
DC 11033	124	279	117	263	3	22	4.7 ± 0.00	7.8 ± 0.34	0	2	0	1	
DC 11040	99	232	92	206	5	12	4.5 ± 1.42	5.2 ± 0.31	0	1	0	1	
DC 11125	311	175	288	167	19	14	7.0 ± 2.00	8.2 ± 0.70	1	1	1	1	
DC 11142	164	209	160	203	7	11	4.4 ± 1.00	5.4 ± 0.37	1	4	1	3	
DC 11146	94	188	88	181	10	12	9.7 ± 7.27	6.2 ± 0.48	0	3	0	3	
DC 11164	229	190	224	181	5	4	5.8 ± 2.92	2.1 ± 0.49	0	0	0	0	
DC 11209	181	151	158	134	3	6	4.4 ± 1.68	4.0 ± 0.38	0	3	0	2	
DC 11221	292	124	259	112	31	7	9.7 ± 1.83	5.9 ± 0.49	5	4	5	4	
DC 11244	170	144	157	135	8	9	6.8 ± 1.80	6.3 ± 0.50	0	1	0	1	
POB 14/2013	79	175	70	164	2	13	2.5 ± 0.16	7.6 ± 3.54	0	0	0	0	
POB 15/2013	79	178	43	168	5	12	6.3 ± 0.00	6.7 ± 0.42	0	0	0	0	
POB 19/2013	92	170	87	160	6	10	6.2 ± 2.80	6.1 ± 0.33	0	1	0	1	
POB 20/2013	101	89	94	82	3	6	4.5 ± 0.80	6.4 ± 0.37	0	0	0	0	
POB 21/2013	60	150	58	140	8	10	13.3 ± 0.00	6.5 ± 0.32	0	3	0	3	
POB 38/2013	13	73	9	65	1	2	7.7 ± 0.00	3.2 ± 0.72	0	0	0	0	
STH 2.3602	253	175	242	170	18	13	9.2 ± 1.70	7.4 ± 0.80	3	2	1	2	
STH 2.3618	240	153	218	140	12	10	6.2 ± 1.40	6.3 ± 0.53	4	3	2	1	
STH 2.3619	82	206	79	196	14	19	20.2 ± 2.14	9.1 ± 0.43	0	4	0	4	
STH 2.3621	182	214	180	207	21	13	11.5 ± 2.40	6.1 ± 0.36	2	1	1	1	
STH 2.3626	284	153	234	146	12	10	9.1 ± 1.66	6.5 ± 0.46	3	2	3	2	
STH 2.3628	293	155	269	149	20	14	8.3 ± 2.10	9.1 ± 0.70	0	1	0	0	
STH 2.3644	176	112	165	109	7	6	5.8 ± 0.80	5.7 ± 0.43	1	3	1	3	
STH 2.3646	243	80	215	75	7	6	4.8 ± 0.97	7.1 ± 1.17	0	1	0	1	
STH 2.3659	164	134	161	124	8	4	5.2 ± 0.98	2.7 ± 0.35	0	0	0	0	
STH 2.3668	215	128	200	123	17	14	8.8 ± 1.40	10.9 ± 0.77	1	2	1	2	
STH 2.9156	36	139	36	122	1	5	2.7 ± 0.00	3.4 ± 0.30	0	0	0	0	
Total/average*	4894	4571	4495	4317	319	300	*8.2 ± 1.62	*6.5 ± 0.63	27	49	22	43	

All the haploid plants came from ten of the tested genotypes (DC 11033, DC 11040, DC 11146, DC 11209, DC 11244, POB 19/2013, POB 21/2013, STH 2.3619, STH 2.3628 and STH 2.3646) treated with the higher 2,4-D concentration (Fig. 4). In turn, among the remaining genotypes, 100 mg/L 2,4-D treatment resulted in not less than one third of haploid plants (as in genotype STH 2.3621), while for two genotypes (DC 11142 and STH 2.3644), haploids were obtained in almost 80%. It is worth emphasizing that there was no genotype that produced haploids only after treatment with the lower auxin concentration.

The obtained haploid plants were subsequently transferred from Petri dishes to perlite and next to soil with sand (Fig. 3H). The next critical step in DH production is the chromosome doubling procedure, including colchicine treatment. As not all plants survived it, some changes in the percent of the obtained DH plants in different 2,4-D concentrations were found compared to haploid plants (Figs. 3I, 5). This procedure turned out to be an essential step for many of the studied genotypes, because, not all of the plants survived (Table 2). The highest plant mortality at 50 mg/L 2,4-D was observed in genotypes STH 2.3602, STH 2.3618, STH 2.3621, where at least half of the haploids did not survive colchicine treatment. In some genotypes (DC 11003, DC 11021, DC 11027, DC 11125, DC 11221, STH 2.3626, and STH 2.3644), all obtained haploid seedlings continued their growth (27% of lines). On the other hand, 100 mg/L 2,4-D treatment led to the death of the obtained plants in five genotypes (DC 11033, DC 11142, DC 11209, STH 2.3618 and STH 2.3628). However, higher auxin concentration contributed to 100% survival in as many as 17 genotypes i.e., about 58% of all lines. Attention should be drawn to genotype DC 11221, for which the highest number of haploid seedlings was obtained after 50 mg/L 2,4-D treatment, and where all of plants have successfully undergone colchicine treatment. Both applied auxin treatments led to the death of plants of genotypes DC 11164, POB 14/2013, POB 15/2013, POB 20/2013, POB 38/2013, STH 2.3659 and STH 2.9156.

Figure 4 Percent of haploid plants [%] grown on MS medium obtained after application of two tested 2,4-D concentrations: 50 mg/L –blue bars or 100 mg/L –grey bars.

Bars show what percentage of the obtained plants came from a given treatment (50 mg/L or 100 mg/L 2,4-D).

Figure 5 Percent of plants [%] after colchicine treatment depending on two tested 2,4-D concentrations: 50 mg/L –blue bars or 100 mg/L –grey bars.

Bars show what percentage of the obtained plants came from a given treatment (50 mg/L or 100 mg/L 2,4-D).

Effect of 2,4-D concentration on DH line development and seed production

The number of obtained DH lines and seeds is presented in Table 3 and Fig. 6. The results of the experiment clearly showed the advantage of treatment with 100 mg/L 2,4-D, which in general allowed to obtain twice as many DH plants (44 plants) compared to 50 mg/L 2,4-D (22 plants) (Fig. 3J). This tendency was observed in 15 genotypes (DC 11003, DC 11033 (Fig. 3K), DC 11040, DC 11142, DC 11146, DC 11209, DC 11221, DC 11244, POB 19/2013 (Fig. 3L), POB 21/2013, STH 2.3619 (Fig. 3M), STH 2.3628, STH 2.3644, STH 2.3646 and STH 2.3668), which accounted for 52% of all tested genotypes. However, there were 3 genotypes (DC 11021, STH 2.3602 and STH 2.3621) where a lower 2,4-D concentration resulted in a higher number of DH lines; other 4 genotypes (DC 11027, DC 11125, STH 2.3618 and STH 2.3626) formed the same number of DH line, regardless of the applied auxin dose. The highest number of DH lines (4 DH lines) developed from genotypes DC 11003, DC 11142, DC 11221, and STH 2.3619 after 100 mg/L 2,4-D. Therefore, these results confirmed better efficiency of the higher auxin concentration and strong dependence of the method effectiveness on plant genotype.

Table 3 Effect of 2,4-D concentration (50 mg/L or 100 mg/L) on the development of doubled haploid plants (DH lines) and seed production (Number of seeds, Seeds/DH line).

Genotype	DH lines	Number of seeds	Seeds/DH line	
	50 mg/L	100 mg/L	50 mg/L	100 mg/L	50 mg/L	100 mg/L	
DC 11003	2	4	0	120	0	30	
DC 11021	3	1	100	18	33	18	
DC 11027	1	1	36	40	36	40	
DC 11033	0	2	0	5	0	3	
DC 11040	0	1	0	6	0	6	
DC 11125	1	1	6	35	6	35	
DC 11142	1	4	0	24	0	6	
DC 11146	0	2	0	91	0	46	
DC 11164	0	0	0	0	0	0	
DC 11209	0	3	0	68	0	23	
DC 11221	3	4	87	85	29	21	
DC 11244	0	1	0	91	0	91	
POB 14/2013	0	0	0	0	0	0	
POB 15/2013	0	0	0	0	0	0	
POB 19/2013	0	1	0	0	0	0	
POB 20/2013	0	0	0	0	0	0	
POB 21/2013	0	1	0	207	0	207	
POB 38/2013	0	0	0	0	0	0	
STH 2.3602	3	1	114	0	38	0	
STH 2.3618	3	3	278	54	93	18	
STH 2.3619	0	4	0	774	0	194	
STH 2.3621	2	1	82	0	41	0	
STH 2.3626	2	2	69	291	35	146	
STH 2.3628	0	1	0	0	0	0	
STH 2.3644	0	3	0	121	0	40	
STH 2.3646	0	1	0	169	0	169	
STH 2.3659	0	0	0	0	0	0	
STH 2.3668	1	2	0	8	0	4	
STH 2.9156	0	0	0	0	0	0	
Total	22	44	772	2207	311	1097	

Figure 6 Percent of DH lines [%] obtained after application of two tested 2,4-D concentrations: 50 mg/L –blue bars or 100 mg/L –grey bars.

Bars show what percentage of the obtained plants came from a given treatment (50 mg/L or 100 mg/L 2,4-D).

Although the number of DH lines obtained is undoubtedly the most important indicator of the effectiveness of the wide crossing method, plant fertility, and thus the ability to produce seeds is another very important factor. In total, DH lines produced 2,979 seeds of which 74% (2,207) were derived from plants treated with 100 mg/L 2,4-D. The most productive genotypes were: POB 21/2013, STH 2.3619 and STH 2.3646 (207, 194, and 169 seeds per DH line, respectively) after 100 mg/L 2,4-D treatment, while 50 mg/L 2,4-D did not allow to obtain more than 100 seeds per DH line. In addition, the higher auxin concentration was associated with seed formation in 62% of genotypes, while a lower dose only in 28% of all tested genotypes.

Discussion

Various biotechnological techniques are used to obtain DH, which are indisputably a powerful tool in plant breeding. In oat (Avena sativa L.), wide hybridization through interspecific crosses is the most commonly applied DH system (Rines, 2003; Sidhu et al., 2006; Ishii et al., 2013; Marcińska et al., 2013; Nowakowska et al., 2015; Warchoł et al., 2018), and wide crossing with maize (Zea mays L.) is most widely used in the commercial production of oat DH lines (Skrzypek et al., 2016). Many factors have been shown to affect the efficiency of haploid/DH production, and unfortunately current methodology is still not cost-effective for a large-scale implementation.

As mentioned before, fertilization with alien pollen causes low embryo viability and zygote abortion during the initial stages of embryonic development. Post pollination application of PGRs, as a key step in chromosome elimination, can increase the recovery of haploid embryos to a state suitable for plant growth on nutrient media (Mahato & Chaudhary, 2019). Wedzony et al. (1998) found that dicamba and picloram were the most effective treatments for ovary growth and embryo production in triticale. Likewise, Knox, Clarke & De Pauw (2000) reported a similar effect of dicamba in durum wheat. However, 2,4-D has been indicated as the most effective in converting embryos to haploid plants and DH development in oat pollinated with maize (Marcińska et al., 2013; Wedzony et al., 2015; Warchoł et al., 2018). Many years of research have prompted us to examine two concentrations of this synthetic auxin in the context of DH production efficiency, taking into account its influence on each step of this complex process. Chaudhary et al. (2015) and Mahato & Chaudhary (2015) showed in durum wheat × I. cylindrica that haploid induction was most responsive at a concentration of 250 mg/L 2,4-D. The application of 100 mg/L 2,4-D in oat haploid production led to the production of 1,5–2% embryos per floret, depending on the studied genotype (Sidhu et al., 2006). Marcińska et al. (2013) demonstrated that this parameter in most of the studied genotypes did not exceed 10%, and 24% embryos per florets were obtained only in one genotype. In turn, Warchoł et al. (2018) tested 33 oat genotypes that did not produce more than 9 embryos per hundred emasculated florets. In our study, significantly higher values of this parameter were obtained that amounted to 26% at 50 mg/L 2,4-D, while at 100 mg/L, they were not higher than 11%. However, most of the conducted studies did not consider the effect of auxin on further stages of DH production. Our findings have indicated that the influence of auxin should not be estimated/based on a single parameter. It was clearly shown that the haploid-embryo-per-floret ratio did not directly reflect the number of haploid/DH plants. Moreover, we demonstrated that the number of obtained haploids grown on MS medium was higher at 100 mg/L 2,4-D, because after this treatment embryos from over 70% genotypes regenerated into plants, while at 50 mg/L only 40%. In addition, according to Rines (2003), this was a critical step since the germination rate of oat haploid embryos was typically low and did not exceed 20%. Rines et al. (1997) showed that the application of a lower 2,4-D concentration (10 mg dm−3) resulted in obtaining caryopses more normal in shape and color than after a higher auxin concentration. On the other hand, the authors reported that using 100 mg/L 2,4-D contributed to enhanced development of the endosperm and whole embryos. After the application of 100 mg/L 2,4-D, Warchoł et al. (2018) obtained in total 104 haploid plants on MS medium. What’s more, according to Sourour, Olfa & Hajer (2011) the best effect of 2,4-D concentration in durum wheat × maize crosses was obtained with 100 mg/L 2,4-D in developed ovaries (65.90%), embryos (23.62%) and haploid plants (19.79%). The same observation was reported by García-Llamas, Martín & Ballesteros (2004) which showed that treatment with 100 mg/L 2,4-D significantly increased the production of embryos and haploid plants and this concentration of 2,4-D was optimal for bread wheat × maize cross. Ushiyama & Kuwabara (2006) suggested that treatment with 2,4-D at 100 mg/L was also effective for haploid wheat production by Hordeum bulbosum method. 2,4-D in concentration 125, 150, 175 mg/L was used by Sourour, Olfa & Hajer (2011) and in 1000 mg/L by Suenaga (1994). Caryopsis development in these studies was slightly improved at the higher 2,4-D concentration compared with 100 mg/L but the efficiency of embryo formation was four time decreased. Probably the higher concentrations of 2,4-D are toxic for generative tissues.

In our experiment, the tested genotypes produced in total 49 haploid plants after 100 mg/L treatment and 27 haploid plants after 50 mg/L 2,4-D. Moreover, the higher concentration of auxin also appeared to promote the acclimatization of haploids in soil as well as colchicine treatment resistance, which are both the next key steps in DH production. These observations were also confirmed by previous studies. Warchoł et al. (2018) showed that in total 44 DH lines were obtained from the tested genotypes at a higher auxin concentration, while in our study it was 44 DH lines at 100 mg/L and 22 DH line at 50 mg/L 2,4-D. The presented results indicated an additional significant feature of DH production –a high influence of the genotype, manifested in different responses of individual genotypes. This fact is an obstacle to the development of an effective method that could be commercially incorporated into breeding programs.

Conclusions

The efficiency of obtaining oat haploid and doubled haploid plants is strongly influenced by auxin treatment. The number of haploid embryos was comparable after both auxin treatments and did not guarantee their effective development into vigorous plants. Despite the genotypic variation, the higher 2,4-D concentration seemed to be more efficient in the context of obtaining haploid/DH plants as well as their vitality and fertility. Nevertheless, further research should focus on the manipulation of PGR treatments and embryo rescue conditions to optimize the efficiency of the oat DH production technique.

Supplemental Information

Supplemental Information 1 Raw data

Click here for additional data file.

The authors thank Danko Plant Breeding Ltd., Małopolska Plant Breeding Ltd. HBP Polanowice and Plant Breeding Strzelce Ltd., PBAI Group for oat seeds used this study.

Additional Information and Declarations

Competing Interests

Author Contributions

Data Availability

The authors declare there are no competing interests.

Katarzyna Juzoń conceived and designed the experiments, performed the experiments, analyzed the data, prepared figures and/or tables, authored or reviewed drafts of the paper, and approved the final draft.

Marzena Warchoł, Kinga Dziurka, Ilona Mieczysława Czyczyło-Mysza and Izabela Marcińska performed the experiments, authored or reviewed drafts of the paper, and approved the final draft.

Edyta Skrzypek conceived and designed the experiments, performed the experiments, authored or reviewed drafts of the paper, and approved the final draft.

The following information was supplied regarding data availability:

The raw data is available in the Supplemental File.

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
