# Peer review of "The effect of 2,4-dichlorophenoxyacetic acid on the production of oat (Avena sativa L.) doubled haploid lines through wide hybridization"

_PeerJ, doi:10.7717/peerj.12854_

## Round 0.1 · original submission · Minor Revisions

Dear Dr. Skrzypek,

Please address the reviewer comments.

Reviewer 1 ·

Basic reporting

1. English language of the manuscript could be improved. To provide some examples from the text where comprehension may be difficult – lines 167-168, line 183, line 190. I suggest using professional editing service to polish the language.

2. The introduction provides sufficient background to show previous research and to explain the aim of the presented study. The literature is relevant for the field of study. My only suggestion is adding some information (if known) about the possible explanations of the fact that oat is recalcitrant to haploidization.

3. The tables and figures are for relevant. Figures are of high quality. Tables are well designed and allow easily to find relevant results. All figures and tables are properly referred in the text. The captions of tables 2-4 and the captions of figures 4-6 should be completed. In tables 2-4 it should be explained which values are numbers and which are percentages. Moreover, in table 6 it is not clear what Authors would like to show by calculating “seeds/DH lines” (should be number of DH lines). Is it better to obtain more seeds from less DH lines or less seeds from more DH lines and what these values show? This value could be similar for genotypes were low number of seeds were produced by one line and high number of seeds were produced by multiple lines (e.g. DC11003 and DC11027). Moreover, maybe it would be useful to somehow marked “the best” and “the worst” genotypes in tables 2-4. Figures 4-6 the caption should be complemented i.e. the description how the percentages were calculated (and what they really represent) should be added. Could Authors explain why the sum of percentage for 50 mg/L 2,4-D and for 100 mg/L 2,4-D is always 100% (for all genotypes and for all traits)? What really represent the values on figures 4-6? It would be interesting to see how many haploid plants on MS media, successfully acclimatized haploid plants and plant that survive the colchicine treatment there is for a particular genotype in comparison to the number of haploid embryos obtained for a particular genotype. Please use the same colours for 2,4-D concentrations on all charts.

4. Raw data are supplied however some more detailed description should be added.

Experimental design

1. The research is well planned, and the materials and methods are carefully described. The level of details is sufficient to easily repeat the presented experiments. The only information that should be added to materials and methods section are as follows: (i) when (how long after transfer the haploid lines from MS to soil) was the procedure of chromosome doubling performed, (ii) when (how long after the chromosome doubling procedure) was the ploidy level measured, (iii) when the seeds were obtained.

Validity of the findings

No comment

Reviewer 2 ·

Basic reporting

This paper investigates the post-pollination application of different 2,4-D concentrations in the efficiency of oat double haploid production by wide hybridization with maize. The expression of this paper is clear and unambiguous. Literature references and research background were sufficiently provided.
I read this paper with interest. The flow of the experiment seems thorough and the data provided is sufficient, I believe that this paper can be published, but improvements are necessary.

Experimental design

no comment

Validity of the findings

no comment

Additional comments

Below I provide recommendations for improvements.
Line 51-53, a comma should be added before in which.
Line 53-55, a comma should be added before DH technologies
Repeated expression in the caption of figure 2 “(A) control 2n, (B) haploid 1n and (C) doubled haploid 2n.”
I recommend combining your raw data in table 2, 3, 4 to some figures.

Reviewer 3 ·

Basic reporting

The work of Juzoń et al. reveal a new aspect of DH production. The manuscript showed interesting results, however, there are some points for discussion and revision.

Specific comments:
Your introduction needs more detail. I suggest that you improve the description at 2,4-D and auxin to provide more justification for your study (specifically, you should describe the difference between 2,4-D and auxin or IAA et al.). Also introduce the mechanism and advantage of 2,4-D in DH production.

2,4-D is one of the auxin analogues. In your experiment, there was no exogenous auxin treatment. Please consider to use this term more carefully.

More details should to provided in the method of pollination. The time, the temperature and so on.

Experimental design

The authors reported the tested genotypes produced in total 287 haploid plants after 100 mg/L treatment and 27 haploid plants after 50 mg/L 2,4-D. So how about higher concentration 2,4-D treatment, like150mg/L?

Figures 4-6 should be more standardized.

Validity of the findings

no comment

Additional comments

no comment

---

## Round 0.2 · accepted · Accept

The manuscript “The effect of 2,4-dichlorophenoxyacetic acid on the production of oat (Avena sativa L.) 2 doubled haploid lines through wide hybridization” has been reviewed by our reviewers. Doubled haploid plants can contribution to accelerating the breeding process. The authors applied 2,4-dichlorophenoxyacetic acid to improve oat double haploid production by wide hybridization with maize. The study is meaningful, and the reults are inspiring. In the future, how to optimize PGR treatments and embryo rescue conditions is still a need to improve DH production.

The comments from our reviewers have been taken into consideration in the revised version. We are pleased to inform you the paper can be accepted in the present form.
Congratulations!

Reviewer 1 ·

Basic reporting

no comment

Experimental design

no comment

Validity of the findings

no comment

Additional comments

My all previous comments has been taken into consideration. Thank you.

Reviewer 2 ·

Basic reporting

This paper is written with clear and unambiguous, professional English throughout, provided sufficient literature references, field background/context, shared the professional article structure, figures, tables, as well as relevant results to hypotheses.

Experimental design

The experiment was designed within the Aims and Scope of the journal, well defined, relevant & meaningful research question, and sufficient detail & information to replicate.

Validity of the findings

I do not think the raw material is necessary for the paper as described in Tables 2 and 3. These raw data could be provided as supplementary materials.